# The Ying and Yang of Hydrogen Sulfide as a Paracrine/Autocrine Agent in Neurodegeneration: Focus on Amyotrophic Lateral Sclerosis

**DOI:** 10.3390/cells12131691

**Published:** 2023-06-22

**Authors:** Alida Spalloni, Susanna de Stefano, Juliette Gimenez, Viviana Greco, Nicola B. Mercuri, Valerio Chiurchiù, Patrizia Longone

**Affiliations:** 1Laboratory of Molecular Neurobiology, Experimental Neurosciences, IRCCS Fondazione Santa Lucia, 00179 Rome, Italy; destefanosusanna@gmail.com (S.d.S.); j.gimenez@hsantalucia.it (J.G.); p.longone@hsantalucia.it (P.L.); 2Department of Systems Medicine, Università di Roma Tor Vergata, 00133 Rome, Italy; mercurin@med.uniroma2.it; 3Department of Basic Biotechnological Sciences, Intensivological and Perioperative Clinics, Università Cattolica del Sacro Cuore, 00168 Rome, Italy; 4Unity of Chemistry, Biochemistry and Clinical Molecular Biology, Department of Diagnostic and Laboratory Medicine, Fondazione Policlinico Universitario A. Gemelli IRCCS, 00168 Rome, Italy; 5Laboratory of Experimental Neurology, Experimental Neurosciences, IRCCS Fondazione Santa Lucia, 00179 Rome, Italy; 6Institute of Translational Pharmacology, National Research Council (CNR), 00185 Rome, Italy; v.chiurchiu@hsantalucia.it; 7Laboratory of Resolution of Neuroinflammation, Experimental Neurosciences, IRCCS Fondazione Santa Lucia, 00179 Rome, Italy

**Keywords:** hydrogen sulfide, astrocytes, oligodendrocytes, neurodegeneration, motor neuron

## Abstract

Ever since its presence was reported in the brain, the nature and role of hydrogen sulfide (H_2_S) in the Central Nervous System (CNS) have changed. Consequently, H_2_S has been elected as the third gas transmitter, along with carbon monoxide and nitric oxide, and a number of studies have focused on its neuromodulatory and protectant functions in physiological conditions. The research on H_2_S has highlighted its many facets in the periphery and in the CNS, and its role as a double-faced compound, switching from protective to toxic depending on its concentration. In this review, we will focus on the bell-shaped nature of H_2_S as an angiogenic factor and as a molecule released by glial cells (mainly astrocytes) and non-neuronal cells acting on the surrounding environment (paracrine) or on the releasing cells themselves (autocrine). Finally, we will discuss its role in Amyotrophic Lateral Sclerosis, a paradigm of a neurodegenerative disease.

## 1. Introduction

The discovery that the gaseous signaling molecules carbon monoxide, nitric oxide (NO), and hydrogen sulfide (H_2_S) have important roles in cellular physiology opened a new era in biomedical medicine. Hydrogen sulfide is a naturally occurring, flammable, colorless and toxic gas, fatal when inhaled at a high level, due to its poisonous targeting of complex IV of the electron transport chain [1,2]. As the third identified gas transmitter, in the last two decades or so, views of it have drastically changed, recognizing its role as a player in a number of cellular functions in every organ. The enzymatic production of endogenous H_2_S is primarily carried out by three pyridoxal 5′-phosphate (PLP)-dependent enzymes, using L-cysteine as substrates, including cystathionine beta-synthase (CBS), cystathionine gamma-lyase (CSE), and cysteine aminotransferase (CAT) [3,4], in association with the PLP-independent enzyme 3-mercaptopyruvate sulfurtransferase (3MST) (Figure 1). CBS is highly expressed in the brain, with the highest levels in the hippocampus, cerebellum, and cerebral cortex [5], but is also expressed in neural stem cells, where it is involved in their proliferation and differentiation [6,7,8]. Initially believed to be essentially restricted to peripheral organs [9], significant CSE levels have been detected in neuronal tissues [10]. CAT is present in two forms, cytosolic and mitochondrial, and its expression is high in peripheral tissues, as well as in the brain [11]. 3MST is also localized in the neurons and astrocytes of various brain regions [12]. CBS and CSE can also produce H_2_S from homocysteine. Homocysteine, a product of the catabolism of methionine via the trans-sulfuration pathway, is degraded to cysteine that spontaneously releases H_2_S and then is catabolized by brain tissues [13]. The enzymatic pathway of CAT/3-MST is mainly located in neurons inside the mitochondria [14], combines 3MST and CAT, and produces H_2_S from 3-mercaptopyruvate, a product of CAT from L-cysteine and alpha-ketoglutarate.

In 2013, Shibuya et al. [15] reported that H_2_S can also be formed through D-Cysteine via the D-amino-oxidase (DAO), establishing a new H_2_S pathway, situated between the peroxisomes where DAO is situated and the mitochondria, where 3MST is mainly found [16] (Figure 1). In this additional pathway, 3-mercaptopyruvate (3-MP) is generated from D-cysteine by the peroxisomal DAO. The 3-MP is then transferred into the mitochondria, where it is metabolized in H_2_S by 3-MST. It is a PLP-independent pathway, believed to be present only in the brain and kidneys, since DAO has been found, as of now, only in these two organs [15].

Cysteine is a component of tripeptide glutathione (GSH), which is the major antioxidant and the most abundant non-protein thiol in mammals. GSH is believed to be a major source and buffer of cysteine, which is considered one of the rate-limiting factors in the synthesis of GSH, impacting the redox status of the cell [17]. When GSH is not properly metabolized, cysteine is metabolized through two distinct pathways: its oxidative catabolism and the H_2_S pathway.

In the mitochondria, at higher concentrations, H_2_S disrupts the oxidative phosphorylation in the electron transport chain (ETC) during the aerobic metabolism of glucose via the inhibition of cytochrome c oxidase and of the F_0_F_1_-ATP synthase, causing the inhibition of the ETC and a consequential drop in ATP production. At lower concentrations below the toxic levels, it acts as a cytoprotective factor, serving as an electron donor for complex IV [18,19].

The aim of the present review is to discuss the role of H_2_S as a glial autocrine/paracrine-released factor and its contribution to inflammation and to the resolution of the inflammatory responses in aging and neurodegeneration. Finally, we will also discuss the role played by H_2_S in a prototypical example of neurodegenerative disease, i.e., Amyotrophic Lateral Sclerosis (ALS).

## 2. Hydrogen Sulfide, the Vasculature, and the Tripartite Synapse

In the periphery, H_2_S has emerged as one of the regulators of the vascular tone of small arteries, controlling the contraction of smooth muscle cells, the endothelial cells, and the perivascular nerves [20]. 

Blood vessels and astrocytes form the so-called “*neurovascular unit*” that, with the endothelial cells and the microglia, form a bridge between the external environment and the brain parenchyma. With the addition of a neuron surrounded by astrocytic processes, the neurovascular unit forms the so-called “*tripartite synapse*” [21], a structure where astrocytes control neurotransmitter concentrations in the cleft and directly connect neurons and blood vessels through their projections [22]. Furthermore, besides wrapping their thin processes around the neuron, they ramify around the blood vessels, surround them, and form the outermost structure of the blood–brain barrier (BBB) [23]. The astrocytic end-feet processes form a strong association with the pericytes, with the capillary endothelial cells (ECs), and with the perivascular fibroblasts, establishing a network between the neuron and the capillary blood flow. In ECs, H_2_S is mainly generated from cysteine by CSE. 

On one hand, H_2_S has a proangiogenic role per se, affecting angiogenesis directly [24] through its cooperation with NO, since both gases are vasorelaxant [2], or by affecting the vascular endothelial growth factor (VEGF) signaling and being, in turn, regulated by the VEGF. H_2_S is a bona fide endogenous modulator of angiogenesis [25]. On the other hand, H_2_S can also inhibit angiogenesis by reducing the VEGF and HIF-1alpha through the blockage of the STAT3 pathway and via the reduction of the VEGF-mediated CSE expression, attenuating the proliferation of the human umbilical cord blood endothelial cells [26,27]. A reduction in VEGF has been reported as a risk factor in mouse models of ALS and in humans [28]. Recently, elevated plasma levels of sulfides have been associated with cognitive disfunction in Alzheimer’s disease and related dementias (ADRDs), and the current authors have proposed H_2_S as a potential vascular biomarker in ADRDs [29,30]. However, other studies have reported a dysregulation of the reverse trans-sulfuration pathway and sulfhydration in Alzheimer’s disease (AD) and a beneficial effect of H_2_S donors on the AD mouse model 3xTg-AD [31]. Hence, similarly to other scenarios, the bell-shaped control that H_2_S exerts on brain angiogenesis follows the double-edged-sword trend that it has in other districts of the body. Hence, H_2_S controls brain homeostasis endogenously as a product of the glial cells and exogenously as a product of the vasculature. 

Additionally, endogenous H_2_S production is involved in post-ischemic cerebral vasodilation/hyperemia and early BBB disruption after cerebral ischemia, again with a dual effect. While CSE and 3-MST inhibitors are able to significantly prevent early BBB disruption [32], the exogenous administration of H_2_S significantly increases the infarct volume after middle cerebral artery occlusion (MCAO) [33], and high plasma levels of cysteine may lead to poor outcomes in clinical strokes [34]. 

The merging data connecting H_2_S, the vascular system, and ALS will be discussed in the dedicated chapter. 

## 3. Hydrogen Sulfide and Neurons

The knowledge, unfolded in recent years, that H_2_S is a physiologically important neuromodulator, which is particularly abundant in the brain, has paved the way to an exponential search on its role in brain function. In keeping with its two-faced Janus properties [35], H_2_S exerts cytoprotective and/or cytotoxic effects, depending on its concentration [36], via many mechanisms, such as histone modification, DNA methylation, DNA damage and repair, post-translational modification of proteins through sulfur hydration, and as a regulator of autophagy [37].

The neurotoxic face of H_2_S is the “classical” one and often results in respiratory paralysis, probably caused by the inhibition of the brain stem neurons controlling respiration, as well as delirium, vertigo, comas, and even death. It has been and it still is considered a serious occupational hazard, and its toxicology has been extensively reviewed [38,39,40]. 

Between 1989 and 1990, when three independent groups [1,40,41] measured endogenous sulfide levels in animals and in the brain, their observations opened the possibility of considering H_2_S as a neuronal gas transmitter with de facto involvement in the brain’s physiology and metabolism. In the CNS, H_2_S modulates the function of neurons in various brain regions, from the spinal cord to the cortical areas, facilitating, for example, long-term potentiation, or sustaining excitatory postsynaptic potentials [3]. Furthermore, it has been proposed as a neuroprotectant in various neurodegenerative diseases [2].

As a neuromodulator, H_2_S also acts on several ion channels, including ATP-sensitive potassium channels (KATPs) and calcium channels, which are fundamental components of neuronal excitability. In 2021, Dallas et al. [42] identified the voltage-gated K^+^ channel Kv2.1, which is highly expressed in the cortex and hippocampus, where it controls intrinsic neuronal excitability as a target of H_2_S-mediated S-sulfhydration. In primary cortical cultures, H_2_S suppresses synchronous calcium oscillation (SCO) induced by L-type voltage-gated Ca^2+^ and transient receptor potential (TRP) channels [43]. Additional effects of H_2_S on neuronal activity will be further discussed in the “Hydrogen Sulfide and Glial Cells” chapter. 

Furthermore, H_2_S, besides functioning as an inhibitor of the ETC, has recently been described as an ETC substrate of the sulfide:quinone oxidoreductase (SQOR) [44] that, although at low to medium levels, is expressed in brain areas (The human protein atlas https://www.proteinatlas.org/ENSG00000137767-SQOR/tissue, accessed on 3 May 2023). The effects of H_2_S on the ETC have wide metabolic implications, especially for the brain, which is a very metabolically demanding organ, with a distinct cellular specificity of metabolic pathways and neurons that are mainly oxidative, while astrocytes are predominantly glycolytic. In the presence of oxygen, neurons, through mitochondrial activity, process glucose in an oxidative way to yield ATP that helps to maintain synaptic homeostasis and regulate Ca^2+^ concentrations at the neuronal terminals [45]. Under H_2_S treatment, a decrease in ATP and an increase in ADP production have been observed, which is a signal of an increase in aerobic glycolysis [46]. Enhancing glycolysis in neurons has led to a dramatic decrease in glucose utilization in the pentose phosphate pathway, and to increased oxidative stress and apoptosis [47]. Hence, by molding the tightly regulated balance between glycolysis and oxidative phosphorylation in neurons, and between neurons and astrocytes, to meet cellular energetic demands, H_2_S is assisting the maintenance of brain homeostasis. 

## 4. Hydrogen Sulfide and Glial Cells

Important mediators in these physiological and pathophysiological processes are the glial cells: astrocytes, microglia, and oligodendrocytes, which are the most abundant cells, have various structures and functions, and are ubiquitous in all regions of the central nervous system (CNS). 

### 4.1. Astrocytes

Astrocytes support grey matter micro-architecture and compose almost half of the CNS’s volume. They are endowed with ion channels and transport pathways and actively participate in the regulation of the neuronal environment and activity [48]. They assist in several brain functions, from providing trophic support for neurons to controlling extracellular ion and neurotransmitter concentrations and from assisting synapse formation, function, and pruning to preserving the blood–brain barrier [49]. 

Astrocytes, which specifically express glycolytic enzymes, support brain energy by converting 80% of the glucose to pyruvate and lactate through glycolysis [50,51,52], while in neurons, which exclusively express lactate dehydrogenase 1 (LDH1), the conversion of lactate into pyruvate is favored [53]. Furthermore, astrocytes, with a higher NADH/NAD^+^ ratio than neurons, favor the reduction of pyruvate into lactate [54]. They play a key role as homeostatic regulators, as well as in the secretion and metabolism of amino acid-based neurotransmitters. One of their most important homeostatic functions is maintaining the glutamine–glutamate balance. Glutamate, released upon the activation of the excitatory synapse, is then removed from the synaptic cleft, mostly by the action of the astrocytic glutamate transporters. In the astrocyte, the glutamate is then converted to glutamine and shuttled back to the excitatory synapse. The downregulation of the glutamate transporters, which are highly expressed in astrocytes, results in an increased glutamate concentration in the cleft, potentially enhancing neurotoxicity through the overactivation of the glutamate receptors and intracellular Ca^2+^ increase. 

As part of the inflammatory response, astrocyte functions are heterogeneous and dependent on the local inflammatory milieu as well as region- and circuit-specific diversity [55,56]. Inflamed astrocytes release proinflammatory cytokines and chemokines, which are able to enroll peripheral immune cells into the CNS, further activating the astrocytes themselves in a malicious cycle [57]. Two distinct phenotypes of astrocytes have recently been described: A1 and A2 [58], with A1 astrocytes being highly neurotoxic and pro-inflammatory, while A2 astrocytes are neuroprotective and anti-inflammatory. However, the current consensus is that astrocytes may adopt multiple states depending on the context, with only a fraction of common changes being between different states [59]. H_2_S is an astrocyte bioproduct that intervenes in their roles as neuromodulators and inflammatory agents, and possibly in their role as an autocrine mediator that modulates astrocytic phenotypes. By comparing the production of H_2_S in different brain cells, astrocytes are likely to be the main brain producers of H_2_S and there is a possibility that the two different A1 and A2 phenotypes are either regulated by H_2_S or that it might represent a marker for the A2 [60].

Astrocytes regulate their own activities via Ca^2+^ and Na^+^ signals and affect neuronal excitability [48]. Glial cells communicate with their surroundings by increasing intracellular Ca^2+^ concentrations and by propagating the signal as spontaneously occurring Ca^2+^ waves or in response to a variety of stimuli. These regulated Ca^2+^ increases in astrocytes are key in astrocyte–astrocyte and astrocyte–neuron intercommunication. H_2_S affects the glutamate response in neurons, as has been demonstrated in hippocampal slides [5], and induces Ca^2+^ waves in astrocytes [61]. These observations suggest that H_2_S enhances the responses to glutamate in neurons and, probably by provoking the release of Ca^2+^ from the endoplasmic reticulum (ER), shapes Ca^2+^ waves [62]. This hypothesis was confirmed by the observations that thapsigargin, a SERCA inhibitor, and the depletion of Ca^2+^ in the ER were able to suppress the H_2_S-induced Ca^2+^ increase. Moreover, H_2_S inhibited ATP-induced Ca^2+^ increase, suggesting that both H_2_S and ATP release Ca^2+^ from the ER [63,64,65]. The inhibitory effect of H_2_S on ATP-induced Ca^2+^ increase in the absence of extracellular Ca^2+^ was also reported in human vascular endothelial cells [63]. This indicates a strong relationship between the Ca^2+^ increase and the inhibition of ATP-induced Ca^2+^ signals by H_2_S and emphasizes the H_2_S involvement in the control and maintenance of Ca^2+^ in the brain. 

In rat hippocampal astrocytes, H_2_S induces a Ca^2+^ influx, which is reversed by DTT, via the *S*-sulfhydration of the transient receptor sulfhydration A1 (TRPA1) channels [66]. However, in spinal cord astrocytes, Nii et al. [58] reported that H_2_S-induced Ca^2+^ increase was not affected by a TRPA1 antagonist, DTT, removal of extracellular Ca^2+^, or a TRPA1 agonist. Furthermore, a few studies have proposed H_2_S as a modulator of the transient receptor potential vanilloid 1 (TRPV1) [67,68], although others have failed to show a direct interaction between H_2_S and TRPV1 [69,70]. Taken together, these results indicate that TRPA1 and TRPV1 are not involved in the H_2_S-induced Ca^2+^ increase in rat spinal cord astrocytes. Interestingly, a work by Kimura’s group [71] reports that H_2_S is able to increase intracellular Ca^2+^ in astrocytes, showing the morphological characteristics of proinflammatory type-1 (A1), but not in microglia or in neurons [62].

Thus, H_2_S can regulate intercellular communication between astrocytes and the surrounding cells through the modulation of Ca^2+^, although the mechanisms involved in the H_2_S-induced Ca^2+^ responses may vary, depending on the region from which the astrocytes are obtained. Established targets of H_2_S are the *N*-methyl-d-aspartate receptors (NMDARs), which are central in controlling neuronal function, and the T-type calcium channels. On one hand, the H_2_S activation of these receptors facilitates hippocampal long-term potentiation (LTP) [72] and neurite outgrowth [73], in line with its role as a neurotransmitter. On the other hand, excessive facilitation of the NMDARs and T-type calcium channels leads to excitotoxicity and neuronal death, underscoring its two-faced Janus actions [74,75,76]. Hence, H_2_S enhances the responses to glutamate in neurons and induces Ca^2+^ waves in astrocytes, while neuronal activity stimulates CBS expression in spinal cord astrocytes in a paracrine/autocrine mode [77]. 

Voltage-gated Na^+^ is expressed not only in neurons, but also in non-neuronal cells, including astrocyte, as evidenced by in vitro and in vivo studies [78]. H_2_S controls Na^+^ homeostasis in the brain, again in a two-faced manner, with little or no effects at lower physiological levels (<150 microM), while at higher concentrations (>150 microM), it evokes a concentration-dependent disruption of the Na^+^ homeostasis [79], which in turn injures neural function, leading to neuronal death [33,79,80].

Astrocytes are also important pH regulators, a critical function to maintain a proper brain homeostasis [81], with the contribution of H_2_S via its regulation of the Cl^−^/HCO_3_^−^ and Na^+^/H^+^ exchangers [82]. However, similarly to the other beneficial functions, when the production of H_2_S increases to a relatively high levels, it may lead to a drastic decrease in intracellular pH and, therefore, produce harmful effects. Therefore, at physiological concentrations, H_2_S has protective effects, but at higher concentrations, such as those that occur in strokes [33], it may aggravate brain damage by also causing severe intracellular acidification.

Thus, H_2_S fine-tunes several cellular pathways, acting as a paracrine factor released by astrocytes and affecting the surrounding neurons and as an autocrine factor by modulating the astrocytes themselves. 

### 4.2. Microglia

Microglia are the main inflammatory response cells in the central nervous system and the resident macrophages of the CNS. During development, they migrate into the developing neuronal tube from the embryonic yolk sac, inhabiting the brain parenchyma [83]. In adulthood, they do not repopulate from the bone-marrow-derived myeloid (BMD) precursors and are independent from the blood-derived monocytes [84]. Resident microglia can be distinguished from peripheral mononuclear cells/macrophages by unique markers, including the tumor necrosis factor receptor superfamily (Tnfrsf), P2Y purinoceptor 12 (P2Y12), transmembrane protein 119 precursor (TMEM119), CX3C chemokine receptor 1 (CX3CR1), sialic acid binding Ig-like lectin H (Siglech), and the microRNAs miR-99a, miR-125-5p, and miR-342-3p, which are highly expressed by mouse and human microglia [85,86,87]. 

Although microglia have previously been classified as M1 (proinflammatory) and M2 (anti-inflammatory), it is now clear that microglia are more complex than this simple dichotomy, as they often demonstrate characteristics of both phenotypes [88]. Recent works, for example, have identified a microglia subtype called “dark microglia” with ramified and thin processes and prominent staining for IBA1, CD11b, and microglia-specific 4D4, as well as TREM2 that associates with chronic stress, aging, and AD [89]. The inflammatory response is a feature that is consistently described in neurodegenerative diseases [90]. Microglia induce and participate to the inflammatory response, as shown by the upregulation of mRNA encoding for pro- and anti-inflammatory mRNAs and their subsequent release [91]. Under pathological conditions, the inflammatory response, jointly with cell death pathways and oxidative stress, can set in motion a local auto-amplification loop to generate a vicious cycle that further amplifies cell death, oxidative stress, and inflammation [92,93]. H_2_S is one of the factors released by the microglia during their activation and has multifaceted roles [94,95]. 

A few studies have reported that H_2_S acts as a pro-inflammatory mediator by up-regulating the production of cytokines and chemokine via the NF-κB pathway [96], while its anti-inflammatory action has been demonstrated time and again. 

Hu et al. suggested that H_2_S attenuates an LPS-induced inflammatory response in cultured microglia through the regulation of the mitogen-activated protein kinase (MAPK) pathway, and by inhibiting the secretion of the tumor necrosis factor α [97]. H_2_S has been reported to inhibit the pro-inflammatory response in amyloid-β-exposed microglia, as well as in murine models of AD, and attenuated cognitive impairment in a rotenone-rat model of PD, while promoting the microglia polarization from M1 to M2 in the hippocampus [95,98,99]. In traumatic brain injuries, ATB-346 (2-(6-methoxynapthalen- 2-yl)-propionic acid 4-thiocarbamoyl-phenyl ester), an H_2_S-releasing cyclooxygenase inhibitor, showed the ability to reduce secondary inflammation and brain injury [100].

Therefore, the bell-shaped characteristic of the exposure-dependent effect of H_2_S on cellular homeostasis can be applied to astrocytes, while for the microglia, the evidence suggests a clearer role as an anti-inflammatory agent. 

### 4.3. Oligodendrocytes

Oligodendrocytes (OLs) are responsible for the deposition of myelin during development. They are the chief cellular constituent of myelin, and as such, they play an important role in axonal conductance. The generation of the myelin sheaths determines a large cellular expansion that requires a substantial energy demand [101,102] in terms of ATP consumption [103,104]. Lactate, the end product of glycolysis, retains a central role in the generation and maintenance of OLs’ energy demands [105], and, secreted from the OLs via the monocarboxylate transporters (MCTs), it is shuttled to the underlying axons to support their energy demands [106]. Oligodendrocytes are highly vulnerable to glutamate excitotoxicity and oxidative stress [107,108,109]. Their vulnerability to glutamate excitotoxicity is mediated by the activation of glutamate receptors and by a reversal of the glutamate-cysteine exchanger, resulting in cysteine efflux and consequent glutathione (GSH) depletion. A pathway that is independent from the glutamate receptor-induced cell death and that triggers a concurrent increase in oxidative stress [110,111]. 

The cystine-glutamate antiporter, which is called system x_c_^−^ and is a heterodimeric protein complex consisting of a catalytic L chain (xCT) and a regulatory H chain (4F2hc), is a source of glutamate by exchanging extracellular cysteine for intracellular glutamate [112]. Cystine is a precursor for H_2_S and for the synthesis of reduced glutathione (GSH). During glutamate-induced oxidative stress, the levels of cellular cystine and GSH decline in association with the increase in reactive oxygen species (ROS) and deleterious changes in the cellular oxidant/antioxidant balance [113,114]. Furthermore, H_2_S decreases the level of GSH and increases the levels of oxidized glutathione (GSSG), causing a reduction in one of the most important intracellular antioxidant pathways [115]. Hence, we may infer that an increased concentration of H_2_S, reaching toxic levels, could be deleterious for oligodendrocytes for two concurring reasons: the activation of glutamate receptors, particularly NMDAR [80], mediating an increase in Ca^2+^ [113], and the decreased levels of GSH leading to increased levels of intracellular reactive oxygen species (ROS) [116].

MCTs are a family of bidirectional membrane transporters that allow the passage of molecules such as lactate, pyruvate, and ketone bodies [117]. Astrocyte or oligodendrocyte-derived intracellular lactate can exit the cell through MCT1 and/or MCT4. Neurons take up the extracellular lactate through MCT2 [118].

At physiological concentrations in the mitochondria, the sulfide oxidation pathway converts H_2_S to the largely innocuous products thiosulfate and sulfate [94], and by stimulating intracellular Ca^2+^ increase, it supports the lactate supply to the neurons from the astrocytes, helping to maintain energy homeostasis [119] (Figure 2A). The increased glycolytic flux is balanced by the increased production of lactate, generating a redox-neutral cycle that prevents the building of cytosolic NADH and supports the continuous production of ATP. Lactate is a critical energy substrate for axons, especially long axons (such as the ones of the motor neurons), maintaining the metabolic needs of intracellular transport and signal transduction. We may envision a condition in which H_2_S assists cell metabolism and homeostasis by filling in for oxygen in the mitochondrial respiratory chain and nurturing the cell by boosting lactate fluxes (Figure 2A).

At higher concentrations or under prolonged exposure, H_2_S turns poisonous to the mitochondrial respiratory chain, leading to the obstruction of ATP production and excessive ROS production. Furthermore, too much lactate can promote further H_2_S production [120], triggering a deadly cycle. Moreover, Tsugane et al., 2007 showed that differentiated astrocytes acquire sensitivity to H_2_S that is diminished by their transformation into reactive astrocytes, possibly causing a further increase in H_2_S in the cleft (Figure 2B) [71]. 

Zhang et al. [121] have recently demonstrated that in a condition of persistent hypoxia, MCT1 function is inhibited, leading to intracellular lactate accumulation and acidosis in oligodendrocytes. Excessive or continuous exposure to H_2_S may also inhibit MCT1 in the OLs, which could disrupt lactate transport in and out of the OLs, resulting in a further impairment of their supporting function to the neurons’ wellbeing (Figure 2B).

## 5. Hydrogen Sulfide and the Resolution of Inflammation

In consistency with the previously described protective and beneficial effects of H_2_S, this gaseous molecule is now recognized as a new member of the of the resolution of inflammation mediators. 

Initially, the resolution of inflammation was believed to essentially be a passive process caused by the cessation or dilution of inflammatory mediators. However, it now represents a novel paradigm in biology, whereby it is rather an active process, initiated during the peak of acute inflammation with the aim of reducing the inflammatory process and ultimately promoting tissue repair and preventing the transition into chronic inflammation [122]. 

The resolution of inflammation is a process, governed by the production and release of pro-resolving lipid mediators in a well-controlled spatial and temporal process by the very same cells that have been recruited to the inflamed site and that previously produced pro-inflammatory mediators. These cells then undergo a metabolic switch and begin to produce pro-resolving lipids, such as the resolvins, protectins, and maresins that are derived from the omega 3 polyunsaturated fatty acids EPA and DHA [123]. These lipids act as “immunoresolvents”, since they inhibit the infiltration of leukocytes in the inflammatory site while inducing the recruitment of non-phlogistic mononuclear cells. They promote the killing and clearance of pathogens and the macrophage-mediated phagocytosis of debris and apoptotic cells (efferocytosis), as well as inhibiting the production of proinflammatory cytokines and inducing that of anti-inflammatory mediators and promoting tissue regeneration [124,125]. Overall, they actively terminate inflammation and drive the restoration of full tissue homeostasis by activating the cardinal signs of resolution: removal, relief, restoration, regeneration, and remission [126].

These pro-resolving mediators are now considered a family of a bigger superfamily of pro-resolving agents, which not only include lipids but also other classes of molecules, such as peptides (e.g., annexin 1, α-MSH, chemerin, and galectin-1), autacoids (e.g., adenosine), and gases [127,128]. Interestingly, H_2_S has a positive interlink with annexin A1, another pro-resolving mediator that is regulated by glucocorticoids [129], whereby the two mediators form a solid association both in vitro and in vivo via a triggering action of H_2_S on the release of AnxA1 that, in turn, activates nonredundant pro-resolving pathways [130]. Furthermore, H_2_S can induce the signature processes of resolution of inflammation, such as efferocytosis [131], inhibition of leukocyte–endothelial cell adhesion and edema formation [132], and macrophage shift towards a pro-resolving and M2-like phenotype. Of note, since H_2_S has been shown to promote healing of ulcers and the resolution of mucosal inflammation in the gastrointestinal tract, and since enteric bacteria are a significant source of H_2_S, enterocytes and colonocytes can represent a metabolic barrier to the diffusion of bacteria-derived H_2_S, not only into the subepithelial space but also in current and novel understandings of the gut–brain axis [133]. 

Lastly, the metabolism of H_2_S is also strictly connected with a novel class of pro-resolving mediators that are derived from the conjugation of cysteine-containing glutathione (GSH) with either maresins, protectins, or resolvins to yield the so-called “conjugates in tissue regeneration” that include MCTRs, PCTRs, and RCTRs, which all share in common the ability to promote tissue regeneration and resolution of infections in several model organisms [134].

## 6. Hydrogen Sulfide and Amyotrophic Lateral Sclerosis

The current literature on H_2_S has a wealth of information suggesting its roles as a neuroprotectant and druggable target in neurodegenerative diseases. We dedicate this chapter to the potential beneficial/harmful roles that H_2_S might have in Amyotrophic Lateral Sclerosis (ALS), a disease that we have been studying for more than two decades. 

Alteration of both homocysteine and GSH, both related to the H_2_S pathways, have been linked to the occurrence of ALS. 

### 6.1. Homocysteine

High median levels of the sulfur-containing amino acid homocysteine (a product of the methionine metabolism) have been reported in the plasma of ALS subjects and in animal models of ALS (mainly the SOD-1-linked models). Although the motive for the higher homocysteine levels has not been fully grasped, the increased levels have been linked to a greater risk of the development and progression of ALS-linked motor neuron degeneration via the promotion of oxidative stress and the disruption of Ca^2+^ homeostasis, leading to excitotoxicity [135,136,137,138]. The higher homocysteine levels have mainly been measured in the liquor and have eventually been linked to the integrity of the brain–blood barrier [32]. Hence, on one hand, the detoxifying action of CBS, by decreasing homocysteine levels and producing H_2_S with antioxidant functions, can be interpreted as protective. On the other hand, we may hypothesize a context where the excessive detoxifying action of CBS leads, in the long run, to an overproduction of H2S. A few years ago, methylcobalamin (vitamin B-12), a vitamin able to reduce homocysteine and H_2_S, was proposed as a potential ALS treatment, reaching phase II/III clinical trials (NCT00444613 and NCT00445172). In the study by Kaji et al. [139], the authors concluded that ultra-high doses of methylcobalamin, although not significantly superior compared to the placebo, may improve the prognosis of ALS if administered in the disease’s early stages. In 2022, the Japanese Early-Stage Trial of Ultrahigh-Dose Methylcobalamin for ALS (JETALS) consortium [140] published the results of a phase 3 clinical trial and concluded that ultrahigh doses of methylcobalamin are safe and are able to significantly slow down the clinical progression of the disease in patients at the early stage and with moderate progression. 

### 6.2. Glutathione (GSH)

Uncontrolled free radicals are a source of damage for all cellular components. Hence, a dysfunctional antioxidant defense mechanism could result in a state of oxidative stress, leading to the cellular deterioration in the long run. The tripeptide GSH is the most abundant thiol molecule found in tissue, with particularly high concentrations in neurons [141]. Glutathione in its reduced form (GSH), reacting in a non-enzymatic way with reactive oxygen species (ROS), scavenges free radicals. GSH is oxidized to glutathione disulfide (GSSG) by the action of glutathione peroxidase, and then the GSH is restored from the GSSG by glutathione reductase. A redox imbalance (a decreased GSH/GSSG ratio) or an increased dysfunction in the GSH homeostasis has been linked to the development and progression of ALS [142]. Thus, H_2_S production and metabolism might have an impact on ALS, not only per se but also as a molecule affecting a broader redox status of the cell through an aberrant GSH metabolism.

Switzer et al. [143] have just reported a strong link between the detoxification of H_2_S and superoxide dismutase [Cu-Zn] (SOD1), which they found between the predominantly cytosolic H_2_S detoxifying enzyme, halting H_2_S-mediated cytotoxicity, and the production of reactive sulfur species (RSS). Mutations in the SOD1 gene were the first reported familiar link to ALS; hence, it is feasible to hypothesize a correlation between the familial forms of ALS, linked to SOD1 mutations, and increased levels of H_2_S. 

Considering the growing body of data about the important role that non-cell-autonomous mechanisms have in ALS-related motor neuron demeanors, understanding the contribution of the non-neuronal components of the neurovascular unit could help reshaping the neuron-centric theory of the ALS etiology. In this context, H_2_S, as an important element released by the cellular component of the unit, could play a relevant role. Interestingly, Månberg et al. [144] reported that sporadic ALS patients and two ALS mouse models (SOD1G93A and TARDBPQ331K/Q331K) presented an enrichment of genes related to the vascular and perivascular system at the pre-symptomatic stage (4–6 months for SOD1G93A; 5 months for TARDBPQ331K/Q331K). They found a striking enrichment in two signature genes of the perivascular fibroblasts: secreted phosphoprotein 1 (SPP1), also known as osteopontin (OPN), and the collagen VI-related dystrophy gene A1 (COL6A1). In vascular diseases, OPN is upregulated in acute and chronic inflammation and participates in the modulation of the inflammatory response. H_2_S modulates the mRNA expression of OPN, although its control is still controversial, depending on the organ. It suppresses the nicotine- and LPS-induced downregulation of OPN mRNA in periodontal osteoblasts [145] while ameliorating the vascular calcification by also decreasing the aortic OPN mRNA levels [146]. 

In 2015, we showed toxic levels of H_2_S in the liquor of sporadic Amyotrophic Lateral Sclerosis (ALS) patients [147] and measured high levels of H_2_S in the classic ALS mouse model SOD1G93A. Furthermore, in a follow-up study, we were able to demonstrate that the pharmacological inhibition of H_2_S production, by the systemic dual inhibitors of cystathionine-β-synthase and cystathionine-γ lyase amino-oxyacetic acid (AOA), decreased the content of H_2_S in primary spinal cord cultures. Furthermore, the AOA treatment was able to decrease the amount of H_2_S in the muscle and cerebral tissues of the SOD1G93A mouse and to increase, by approximately ten days, the lifespan of the SOD1G93A female [148]. Similarly, to other neurodegenerative diseases, in ALS, a variety of factors concur to its development and/or progression, such as H_2_S, as well as sex, which probably justifies the partially beneficial AOA data. By using primary spinal cord cultures prepared from the C57BL6 mouse, we also determined its higher toxicity to motor neurons, with respect to the GABAergic neurons. We also proved that it is a factor released by astrocytes and microglia [147], since its concentration in the culture media significantly decreased when we halted the glial cells’ proliferation with arabinoside-C (Ara-C) or we induced inflammation with LPS [147]. A proteomic analysis performed on the primary cultures, treated with H_2_S (200 μM, 18 h), revealed that, under this toxic protocol, H_2_S activates oxidative and cell death pathways, which are mainly the Nrf-2-mediated oxidative stress response and peroxiredoxins. We also showed that in these cultures, the H_2_S-mediated toxicity is halted, at least in part, by necrostatin, a potent necroptosis inhibitor, and by the Bax (Bcl-2-loke protein 4) inhibitor V5 [149]. Interestingly, in 2007, Nagai et al. [150] reported that a Bax-dependent soluble toxic factor(s) released by ALS-mutant astrocytes was toxic to motor neurons. Collectively, our data reflect the view of H_2_S as a pathological factor contributing to ALS-mediated motor neuron death. Could H_2_S be the or a toxic factor facilitating the non-cell autonomous toxicity in ALS and beyond?

## 7. Conclusions and Future Perspectives

One of the challenges presented in the field of H_2_S is its opposite effects on cellular homeostasis and survival, as determined by its fluctuating concentrations. Comprehension of its proper modulation is, therefore, warranted. Glial cells, in particular astrocytes, are the mayor producers of H_2_S in the brain, and sufficient evidence has been collected to corroborate H_2_S’s role as a glial modulator and as a paracrine/autocrine factor that can either help to fight or facilitate the development of neurological diseases. 

The growing appreciation of the importance of H_2_S in bodily physiology and pathology is testified by the growing interest in this gas transmitter and its pathways as druggable targets. 

## Figures and Tables

**Figure 1 cells-12-01691-f001:**
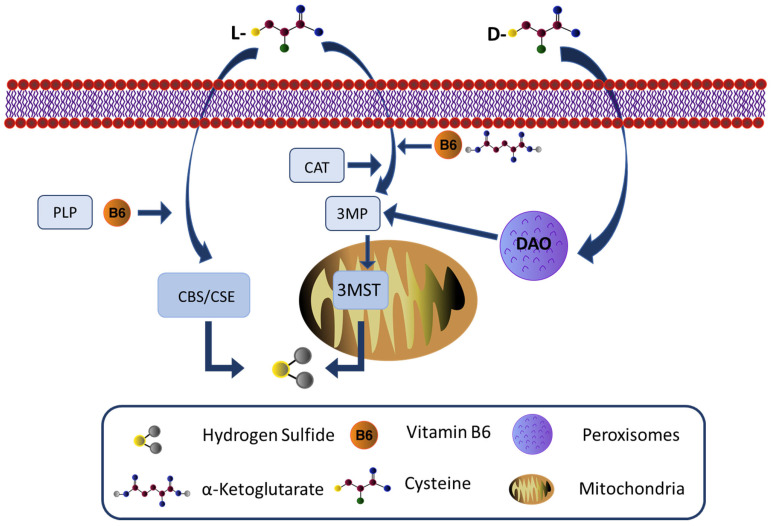
Schematic overview of hydrogen sulfide production. Endogenously, H_2_S is produced by three enzymes: cystathionine-beta-synthase (CBS), cystathionine gamma-lyase (CSE), and cysteine aminotransferase (CAT)/3-mercaptopyruvate sulfurtransferase (3MST). It has also been reported that it can be produced in brain homogenate when D-cysteine is added instead of L-cysteine. CBS, CSE, and CAT enzymes require pyridoxal-5′-phosphate (PLP) as a cofactor and all use L-cysteine as a substrate which is a by-product of L-methionine, homocysteine, and cystathionine. 3-mercaptopyruvate is converted from cysteine by the action of cysteine aminotransferase. As a semi-essential amino acid, cysteine can be obtained from alimentary sources or liberated from the catabolism of endogenous proteins. 3MST uses 3 mercaptopyruvate (3MP), generated by CAT from L-cysteine and a-ketoglutarate (a-KG), in association with vitamin B6 as a substrate to produce H_2_S. Both enzymes have a cytosolic and a mitochondria isoform. The mitochondria subcellular isoform is probably the relevant one, since it is where cysteine is preferentially found. Finally, D-Amino acid oxidase (DAO), which is localized to peroxisomes, can produce 3MP through the oxidative deamination of D-cysteine. 3MST can produce H_2_S along with DAO, exploiting the interaction between those organelles, normally in close proximity and known to exchange metabolites.

**Figure 2 cells-12-01691-f002:**
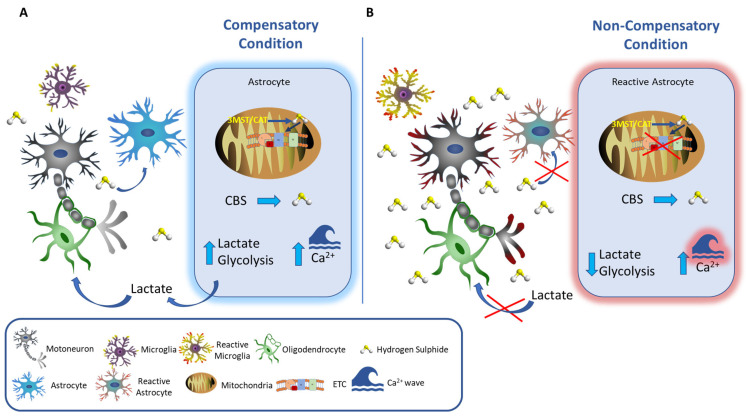
A brief and general overview of hydrogen sulfide as a signaling molecule in brain cellular communication. H_2_S, a byproduct of mammalian tissues through enzymatic and nonenzymatic pathways, is mainly produced in the brain by astrocytes. Compensatory condition (**A**): H_2_S, is metabolized by astrocytes, mainly via cytosolic enzyme CBS (cystathionine β synthase) and via the 3MST/CAT pathway in the mitochondria. In the early phases of brain injuries (hypoxic conditions, pathological degeneration), H_2_S supports the mitochondrial respiratory chain and supports intra-cellular communication by sustaining Ca^2+^ waves while boosting glycolysis and its end product, lactate. Lactate, released by glial cells in the cleft, is captured by the neuron and used as a source of energy. Non-compensatory condition (**B**): Excessive presence of H_2_S (either due to a prolonged exposure or due to levels above the physiological limit) impairs the mitochondrial respiratory chain and leads to a decrease in glycolysis and lactate (as indicated by the red cross), depriving the neurons of their metabolic fuel. Additionally, a non-compensatory condition may well facilitate the astrocytes’ transformation from differentiated to reactive and make them unable to sustain Ca^2+^ waves. Reactive astrocytes have a diminished ability to remove extracellular H_2_S, further increasing its concentration in the cleft.

## Data Availability

No new data created.

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
