# Peer review of "The Ying and Yang of Hydrogen Sulfide as a Paracrine/Autocrine Agent in Neurodegeneration: Focus on Amyotrophic Lateral Sclerosis"

_cells, 2023, doi:10.3390/cells12131691_

Round 1
Reviewer 1 Report
Spalloni et al. review manuscript aims to describe the role of hydrogen sulfide as a paracrine or autocrine agent in neurodegeneration. Alteration of H2S metabolism and neurodegeneration or behavioural abnormalities has been widely studied. The association of H2S metabolism alterations with amyotrophic lateral sclerosis is a relative new area of research and a fascinating field which deserves to be studied. Although the topic of the present manuscript is of foremost interest, the manuscript itself presents many gaps and needs to be further improved.
Major comments:
1) In the introduction paragraph, authors introduce the metabolic pathways leading to hydrogen sulfide generation. Authors should provide more details on the biochemical pathway/be clearer in the explanation.
2) In the paragraph hydrogen sulfide and neurons, lines 154-157, authors refer to hydrogen sulfide as an electron transport chain (ETC) substrate, as previously reported by Goubern et al. (in the present manuscript ref. No. 19). However, the same authors in a follow-up paper (Lagoutte E et al. Oxidation of hydrogen sulfide remains a priority in mammalian cells and causes reverse electron transfer in colonocytes. Biochim Biophys Acta. 2010, 1797(8):1500-11) reported that sulfide oxidation through the ETC was not detected in brain mitochondria. Please discuss.
3) The main novelty of this research review is to describe the link between aberrant H2S metabolism and ALS. However, in the dedicated paragraph, Hydrogen sulfide and Amyotrophic Lateral Sclerosis, the physio-pathological explanation of how this gasotransmitter may be involved in the onset of this disease is poor or contradictory:
- in line 422-441 authors discuss the putative role of higher homocysteine levels on the onset of ALS, concluding that “the detoxifying action of CBS, by decreasing homocysteine levels and producing H2S, with antioxidant functions can be interpreted as protective” which is in sticking contrast with the hypothesis supported by the authors, namely higher H2S levels may be detrimental – and perhaps – linked to ALS onset. Please discuss.
- In the same paragraph, authors do not report any explanation of the mechanism behind the accumulation of homocysteine in ALS patients. For instance, high circulating homocysteine levels is a “trade-mark” of homocystinuria, a disease associated with (loss-of-function) CBS mutation. This results in accumulation of its substrate (homocysteine) and consequent lower levels of its product (H2S). Therefore, detrimental sulfide accumulation is directionally inverse to homocysteine accumulation. Please discuss.
- In line 479-484, authors report that treatment with AOA, an inhibitor of CBS and CSE has been reported to increase the lifespan of SOD1G93A female. However, treatment with AOA is supposed to i) increase homocysteine levels and ii) decrease H2S levels. Increment of homocysteine should exacerbate rather than ameliorate ALS patients’ conditions. Please discuss.
- Why H2S is more cytotoxic towards motor neurons with respect to other neuronal cells (for instance GABAergic neurons)? Please discuss.
- Please clarify how increased levels of H2S (and supposedly increased levels of cysteine through the trans-sulfuration pathway) negatively impacts on glutathione (GSH) biosynthesis.
4) In the text there are misspellings and typos. Please correct them.
Minor comments:
- Figure 1: in the bottom legend within the figuere, words should be written with same font and size. Moreover, the arrow corresponding to the PLP cofactor should put before CAT and not after. Like this seems like authors are proposing that 3MST (and not CAT) is a PLP-dependent enzyme.
- Many times H2S is not subscripted.
- Line 49-51: it is clear that the authors are here describing the role of homocysteine in the bio-generation of hydrogen sulfide, but I would suggest to rephrase the sentence in a clearer way.
- Line 55-58: authors are introducing a recently described pathway, but they are not describing clearly the role of DAO and 3MST in this pathway. I would suggest to rephrase the sentence in a clearer way.
- Line 64: “pyridoxal‐5’‐phosphatase (PLP)‐dependent” – the word “phosphatase” it’s a clear mistake. Phosphatase is an enzyme. I guess the authors meant pyridoxal‐5’‐phosphate (PLP)‐dependent.
- Line 65: saying that the enzymes require vitamin B6 is redundant when enzymes are introduced as PLP dependent enzymes, since PLP is vitamin B6.
- Line 87-90: please add the subject (H2S) at least once in the whole paragraph.
Overall the text is full of mispellings that should be corrected before submitting the paper, even more if it is a re-submission. Please find below some examples:
Line 41: “CBS is highly express in the brain”, change with highly EXPRESSED
Lines 53-55: H2S (change with the number 2 subscript), 3-mercaptopyruvato change to 3-mercaptopyruvate, change to alpha-ketoglutarate.
Line 64: “pyridoxal‐5’‐phosphatase (PLP)‐dependent” – the word “phosphatase” it’s a clear mistake. Phosphatase is an enzyme. I guess the authors meant pyridoxal‐5’‐phosphate (PLP)‐dependent.
Line 66: cystathionina change to cystathionine
Line 71: since “its” where: its in this case stands for IT IS, please change accordingly.
Author Response
We thank the Reviewer for his/her comments, below is a point-by-point response
Spalloni et al. review manuscript aims to describe the role of hydrogen sulfide as a paracrine or autocrine agent in neurodegeneration. Alteration of H2S metabolism and neurodegeneration or behavioural abnormalities has been widely studied. The association of H2S metabolism alterations with amyotrophic lateral sclerosis is a relative new area of research and a fascinating field which deserves to be studied. Although the topic of the present manuscript is of foremost interest, the manuscript itself presents many gaps and needs to be further improved.
Major comments:
1) In the introduction paragraph, authors introduce the metabolic pathways leading to hydrogen sulfide generation. Authors should provide more details on the biochemical pathway/be clearer in the explanation.
As requested by the Reviewer we have implemented the description of the metabolic pathway leading to hydrogen sulfide, providing additional details, also expanding the description of the role of DAO and 3MST as requested in the minor comments.
2) In the paragraph hydrogen sulfide and neurons, lines 154-157, authors refer to hydrogen sulfide as an electron transport chain (ETC) substrate, as previously reported by Goubern et al. (in the present manuscript ref. No. 19). However, the same authors in a follow-up paper (Lagoutte E et al. Oxidation of hydrogen sulfide remains a priority in mammalian cells and causes reverse electron transfer in colonocytes. Biochim Biophys Acta. 2010, 1797(8):1500-11) reported that sulfide oxidation through the ETC was not detected in brain mitochondria. Please discuss.
We thank the Reviewer for the insightful observation. Although the Reviewer is correct, there isn’t an absolute lack of SQOR detection in the brain. The human protein atlas https://www.proteinatlas.org/ENSG00000137767-SQOR/tissue shows a low to medium protein expression in the cerebral cortex, cerebellum, and caudate. Furthermore, Marutani et al. (Marutani E, Sulfide catabolism ameliorates hypoxic brain injury. Nat Commun. 2021 May 25;12(1):3108) reported the ability of isolated brain mitochondria, in the presence of Na2S, of catabolize sulfide supporting the presence of SQOR in brain mitochondria.
3) The main novelty of this research review is to describe the link between aberrant H2S metabolism and ALS. However, in the dedicated paragraph, Hydrogen sulfide and Amyotrophic Lateral Sclerosis, the physio-pathological explanation of how this gasotransmitter may be involved in the onset of this disease is poor or contradictory:
- in line 422-441 authors discuss the putative role of higher homocysteine levels on the onset of ALS, concluding that “the detoxifying action of CBS, by decreasing homocysteine levels and producing H2S, with antioxidant functions can be interpreted as protective” which is in sticking contrast with the hypothesis supported by the authors, namely higher H2S levels may be detrimental – and perhaps – linked to ALS onset. Please discuss.
The Reviewer brings up a good point regarding the contradictory role of Hydrogen sulfide in ALS and its influence in its onset/progression. About the equilibrium between homocysteine and H2S, since homocysteine has been found to increase in ALS, we may reason that the attempt to decrease homocysteine, in the long run, leads to a toxic increase in Hydrogen sulfide levels. We have discussed this hypothesis in the text.
- In the same paragraph, authors do not report any explanation of the mechanism behind the accumulation of homocysteine in ALS patients. For instance, high circulating homocysteine levels is a “trade-mark” of homocystinuria, a disease associated with (loss-of-function) CBS mutation. This results in accumulation of its substrate (homocysteine) and consequent lower levels of its product (H2S). Therefore, detrimental sulfide accumulation is directionally inverse to homocysteine accumulation. Please discuss.
The Reviewer is, indeed, correct in homocystinuria the levels of hydrogen sulfide are low due to mutations in the CBS gene (classic homocystinuria). To the best of our knowledge mutations in CBS of other enzymes linked to Hydrogen sulfide production have not been described in ALS, hence homocysteine increase is not linked to the malfunction of Hydrogen sulfide-linked enzymes. Furthermore, the reason for the increased homocysteine levels in ALS has not been elucidated, it has been observed in patients and animal models (mainly the SOD-1-linked models) and analyzed as cytotoxic factor whit no clear explanation of the why.
- In line 479-484, authors report that treatment with AOA, an inhibitor of CBS and CSE has been reported to increase the lifespan of SOD1G93A female. However, treatment with AOA is supposed to i) increase homocysteine levels and ii) decrease H2S levels. Increment of homocysteine should exacerbate rather than ameliorate ALS patients’ conditions. Please discuss.
We thank the Reviewer for the observation. In the manuscript we have implemented the discussion of our AOA data as requested page 16, lines 14-18 ).
In the AOA paper, we measured the hydrogen sulfide levels but not the homocysteine one, since we focused our attention solely on Hydrogen sulfide. ALS, similarly to other neurodegenerative diseases, is complex and caused by a variety of factors.
- Why H2S is more cytotoxic towards motor neurons with respect to other neuronal cells (for instance GABAergic neurons)? Please discuss.
We do not have a definitive answer to this question. We can put forward a hypothesis about the why. The strongest one is the energy needs of the motor neuron, a highly demanding cell in terms of energy supply. Hence, we may hypothesize that in a prolonged condition in which Hydrogen sulfide retains toxic/poisonous levels due to the redox impairment characterizing ALS mitochondria, the mitochondria succumb to the Hydrogen sulfide toxicity, earlier in a highly vulnerable neuron-like the motor neuron, compared to other types of neurons, like the GABAergic. An additional hypothesis is the effects that prolonged high concentrations of Hydrogen sulfide have, not directly on neurons, but on astrocytes, and how this impact on neurons. Astrocytes have a paramount role in ALS, contributing to the non-cell autonomous degeneration extensively described in ALS. Hence, in prolonged Hydrogen sulfide elevated concentrations, on top of the direct detrimental effects on neurons, through the alterations triggered in astrocytes, Hydrogen sulfide can further affect the demeanor of such a delicate neuron as the motor neuron. In support of this hypothesis, we have already shown that a significant amount of hydrogen sulfide is produced by microglia cells and astrocytes (Davoli A et al., Annals Neur. 2015)
- Please clarify how increased levels of H2S (and supposedly increased levels of cysteine through the trans-sulfuration pathway) negatively impacts on glutathione (GSH) biosynthesis.
Seydi et al., (reference #115) demonstrated that thioacetamide, an H2S donor, by interfering with mitochondrial and lysosomal functions is associated with a reduction in GSH levels. Furthermore, Troung et al. (Truong DH, Eghbal MA, Hindmarsh W, Roth SH, O'Brien PJ. Molecular mechanisms of hydrogen sulfide toxicity. Drug Metab Rev. 2006;38(4):733-44) reported that the incubation of isolated hepatocytes with the H2S donor NaHS results in GSH depletion. They hypothesize that H2S cytotoxicity involving reactive sulfur species depletes GSH and activates oxygen to form ROS.
4) In the text there are misspellings and typos. Please correct them.
We have corrected misspellings and typos.
Minor comments:
- Figure 1: in the bottom legend within the figuere, words should be written with same font and size. Moreover, the arrow corresponding to the PLP cofactor should put before CAT and not after. Like this seems like authors are proposing that 3MST (and not CAT) is a PLP-dependent enzyme.
We have corrected as requested.
- Many times H2S is not subscripted.
Correct
- Line 49-51: it is clear that the authors are here describing the role of homocysteine in the bio-generation of hydrogen sulfide, but I would suggest to rephrase the sentence in a clearer way.
The sentence has been modified.
- Line 55-58: authors are introducing a recently described pathway, but they are not describing clearly the role of DAO and 3MST in this pathway. I would suggest to rephrase the sentence in a clearer way.
As suggested by the Reviewer, the sentence has been modified and implemented.
- Line 64: “pyridoxal‐5’‐phosphatase (PLP)‐dependent” – the word “phosphatase” it’s a clear mistake. Phosphatase is an enzyme. I guess the authors meant pyridoxal‐5’‐phosphate (PLP)‐dependent.
We thank the Reviewer and corrected the mistake.
- Line 65: saying that the enzymes require vitamin B6 is redundant when enzymes are introduced as PLP dependent enzymes, since PLP is vitamin B6.
The phrase has been modified
- Line 87-90: please add the subject (H2S) at least once in the whole paragraph.
As suggested by the Reviewer we have modified the sentences.

Reviewer 2 Report
Although the topic of the review is interesting, the entire work needs to be revised and made fit with the objective proposed in the title.
Paragraphs similar to the one proposed for ALS should be included, where the role of hydrogen sulphide in different neurodegenerative diseases (AD, PD, multiple sclerosis, etc.) is described.
The abstract should be improved. Throughout the article, there are typing errors, a lack or abundance of spacing, as well as parts written in different font sizes and types, suggesting a copy-paste from elsewhere. However, I did not find any plagiarism.
The sentence on lines 440-441 is irrelevant and wrong: the trial outcome mentioned only states that methylcobalamin treatment is safe, but the correlation between vitamin B-12 and the reduction of homocysteine levels was never evaluated. Such a one-sided comment without demonstration is not appropriate.
Good
Author Response
Reviewer #2
We thank the Reviewer for his/her comments, below is a point-by-point response
Although the topic of the review is interesting, the entire work needs to be revised and made fit with the objective proposed in the title.
Paragraphs similar to the one proposed for ALS should be included, where the role of hydrogen sulphide in different neurodegenerative diseases (AD, PD, multiple sclerosis, etc.) is described.
We concur with the Reviewer with regard to the increasing role that H2S holds in neurodegenerative diseases, above all PD and AD, with increasing interest in its role in MS. Following the Reviewer’s suggestion, we have implemented the manuscript with sentences enlightening the role of H2S in additional neuronal and neurodegenerative diseases , other than ALS (page 5, lines 4-8 from the top; page 4-5, lines 1-2 from the bottom, lines 1-3 from the top; page 9; line 6 from the top; page 9, lines 7-8 from the bottom; page 10, the sentence from” Hu et al. suggested” to “and brain injury”; page 11, lanes 4-5 from the bottom). However, this is not the intent of this review, which aims to focus on ALS, as a neurodegenerative disease. Hence, the decision to submit this review for a special issue on ALS, we reason that dedicated chapters addressing the other diseases would be beyond the scope of the present review. To clarify our purpose, we have changed the title to “The Ying and Yang of hydrogen sulfide as a paracrine/autocrine agent in neurodegeneration: focus on Amyotrophic lateral Sclerosis”.
The abstract should be improved. Throughout the article, there are typing errors, a lack or abundance of spacing, as well as parts written in different font sizes and types, suggesting a copy-paste from elsewhere. However, I did not find any plagiarism.
We thank the Reviewer for highlighting these incongruities, and we apologize for them. We went through the whole manuscript, corrected the errors and the typos, and ratified the font.
The sentence on lines 440-441 is irrelevant and wrong: the trial outcome mentioned only states that methylcobalamin treatment is safe, but the correlation between vitamin B-12 and the reduction of homocysteine levels was never evaluated. Such a one-sided comment without demonstration is not appropriate.
We have deleted the sentence.

Reviewer 3 Report
In this study, the authors discuss the role of H2S as a glial autocrine/paracrine released factor and its contribution to inflammation and neurodegeneration, as well as its role in amyotrophic lateral sclerosis. The manuscript is well-organized and clearly written. I have some specific comments:
In Figure 1: The enzymes cystathionine-β-synthase, cystathionine-γ-lyase, and 3-mercaptopyruvate sulfurtransferase are involved in the production of H2S, not its synthesis. Please change the first sentence. (e.g. H2S is produced by three enzymes, cystathionine-β-synthase (CBS), cystathionine-γ-lyase (CSE), and 3-mercaptopyruvate sulfurtransferase (3MST).
In the aim of the study, line 87, Please, point out that you discuss the role of H2S (replace its with H2S)
Line 105: EC cells do not synthesize H2S, but H2S is produced in EC cells (e.g. In endothelial cells, H2S is generated from cysteine by ……….)
Line 205, a put dot after reference
Please write this sentence more clearly because it is not clear how act lipids and what is the role of leukocytes: These lipids act as “immunoresolvents” by inducing cessation of further leukocyte infiltration, stimulation of non-phlogistic mononuclear cells, killing and clearance of pathogens and macrophage-mediated phagocytosis of apoptotic granulocytes (efferocytosis) and cellular debris as well as by inhibiting the production of proinflammatory cytokines while inducing that of anti-inflammatory mediators and promoting tissue regeneration and [120,121].
Figure 2, the 2+ charge of calcium needs to be superscript.
Check the colors and font, in the text, because in some parts of a manuscript, you have different fonts.
Line 251 Death
Write number 2 in the index for each hydrogen-sulfide formula in the manuscript.
Minor editing of English language needed
Author Response
Reviewer #3
We thank the Reviewer for his/her comments, below is a point-by-point response
In this study, the authors discuss the role of H2S as a glial autocrine/paracrine released factor and its contribution to inflammation and neurodegeneration, as well as its role in amyotrophic lateral sclerosis. The manuscript is well-organized and clearly written. I have some specific comments:
In Figure 1: The enzymes cystathionine-β-synthase, cystathionine-γ-lyase, and 3-mercaptopyruvate sulfurtransferase are involved in the production of H2S, not its synthesis. Please change the first sentence. (e.g. H2S is produced by three enzymes, cystathionine-β-synthase (CBS), cystathionine-γ-lyase (CSE), and 3-mercaptopyruvate sulfurtransferase (3MST).
Following the Reviewer’s suggestion, we have modified the first sentence of the figure legend.
In the aim of the study, line 87, Please, point out that you discuss the role of H2S (replace its with H2S)
“Its” has been replaced with “H2S”.
Line 105: EC cells do not synthesize H2S, but H2S is produced in EC cells (e.g. In endothelial cells, H2S is generated from cysteine by ……….)
The line has been corrected.
Line 205, a put dot after reference
Done
Please write this sentence more clearly because it is not clear how act lipids and what is the role of leukocytes: These lipids act as “immunoresolvents” by inducing cessation of further leukocyte infiltration, stimulation of non-phlogistic mononuclear cells, killing and clearance of pathogens and macrophage-mediated phagocytosis of apoptotic granulocytes (efferocytosis) and cellular debris as well as by inhibiting the production of proinflammatory cytokines while inducing that of anti-inflammatory mediators and promoting tissue regeneration and [120,121].
In this sentence, we explained all the roles exerted by specialized pro-resolving lipids.
Since inflammation is caused by the recruitment of leukocytes (granulocytes, monocytes/macrophages, etc) in the damaged tissue, the role of leukocytes is important and, in the sentence, we have listed all the effects that these lipids have on these cells.
We have modified the sentence to make it clearer, but it's essentially a list of the effects they exert on leukocytes and on the typical features of inflammation (recruitment, phagocytosis, efferocytosis, cytokine production, and tissue regeneration)
Figure 2, the 2+ charge of calcium needs to be superscript. Done
Check the colors and font, in the text, because in some parts of a manuscript, you have different fonts.
We thank the Reviewer for the remark. We have edited the manuscript and controlled the font.
Line 251 Death Done
Write number 2 in the index for each hydrogen-sulfide formula in the manuscript. Done

Round 2
Reviewer 1 Report
Following the first revision of the present manuscript, some minor changes should be still done:
- line 43-44: 3MST it is not a PLP-dependent enzyme. Please change
- line 55: 3-mercaptopyuvatO change with 3-mercaptopyuvatE
- line 66: H2S with subscript number 2
- line 84: change redux with redox
- there are some citation reported as plain text in the main text of the manuscript (for instance line 125-134; 137-141; 183-185). please report them as numbers, consistently with other citations
Author Response
Replay to Reviewer #1
Comments and Suggestions for Authors
We thank the Reviewer for his/her comments
Following the first revision of the present manuscript, some minor changes should be still done:
- line 43-44: 3MST it is not a PLP-dependent enzyme. Please change Done
- line 55: 3-mercaptopyuvatO change with 3-mercaptopyuvatE Done
- line 66: H2S with subscript number 2 Done
- line 84: change redux with redox Done
- there are some citation reported as plain text in the main text of the manuscript (for instance line 125-134; 137-141; 183-185). please report them as numbers, consistently with other citations
We thank the Reviewer for pinpointing these inaccuracies that we missed in the first revision, and that now we have corrected them.

Reviewer 2 Report
The authors took the necessary steps to improve the work. The title is now more appropriate to both the special issues and the content of the review.
Some characters that do not conform to the text remain to be fixed (lines 186, 366-375, 378-381).
Author Response
Reviewer #2
Comments and Suggestions for Authors
The authors took the necessary steps to improve the work. The title is now more appropriate to both the special issues and the content of the review.
Some characters that do not conform to the text remain to be fixed (lines 186, 366-375, 378-381).
We thank the Reviewer for his/her suggestions that have surely improved the manuscript.
We have matched the font.

Reviewer 3 Report
The authors accepted all my suggestion and manuscript can be accepted for publication
Author Response
Reviewer #3
Comments and Suggestions for Authors
The authors accepted all my suggestion and manuscript can be accepted for publication
We thank the reviewer for his/her comments which have certainly improved the manuscript.
